# Endocrine and Metabolic Impact of Oral Ingestion of a Carob-Pod-Derived Natural-Syrup-Containing D-Pinitol: Potential Use as a Novel Sweetener in Diabetes

**DOI:** 10.3390/pharmaceutics14081594

**Published:** 2022-07-30

**Authors:** Juan A. Navarro, Juan Decara, Dina Medina-Vera, Ruben Tovar, Antonio J. Lopez-Gambero, Juan Suarez, Francisco Javier Pavón, Antonia Serrano, Marialuisa de Ceglia, Carlos Sanjuan, Yolanda Alfonso Baltasar, Elena Baixeras, Fernando Rodríguez de Fonseca

**Affiliations:** 1Laboratorio de Medicina Regenerativa, Unidad de Gestión Clínica de Salud Mental, Instituto de Investigación Biomédica de Málaga-IBIMA, Hospital Regional Universitario de Málaga, 29010 Málaga, Spain; juan_naga@hotmail.es (J.A.N.); juandecara@uma.es (J.D.); dina.medina@ibima.eu (D.M.-V.); rubentovar7@hotmail.com (R.T.); antonio.lopez@ibima.eu (A.J.L.-G.); juan.suarez@uma.es (J.S.); javier.pavon@ibima.eu (F.J.P.); antonia.serrano@ibima.eu (A.S.); marialuisa.deceglia@ibima.eu (M.d.C.); 2Facultad de Medicina, Campus de Teatinos s/n, Universidad de Málaga, 29010 Málaga, Spain; 3Unidad de Gestión del Corazón, Hospital Universitario Virgen de la Victoria, 29010 Málaga, Spain; 4Facultad de Ciencias, Campus de Teatinos s/n, Universidad de Málaga, 29010 Málaga, Spain; 5Departamento de Anatomía Humana, Medicina Legal e Historia de la Ciencia, Universidad de Málaga, 29010 Málaga, Spain; 6Euronutra S.L. Calle Johannes Kepler, 3, 29590 Málaga, Spain; euronutra@euronutra.eu (C.S.); lab@euronutra.eu (Y.A.B.); 7Departamento de Bioquímica y Biología Molecular, Facultad de Medicina, Universidad de Málaga, 29010 Málaga, Spain

**Keywords:** sweeteners, D-Pinitol, carob fruit, insulin, insulin resistance, liver steatosis, pituitary hormones

## Abstract

The widespread use of added sugars or non-nutritive sweeteners in processed foods is a challenge for addressing the therapeutics of obesity and diabetes. Both types of sweeteners generate health problems, and both are being blamed for multiple complications associated with these prevalent diseases. As an example, fructose is proven to contribute to obesity and liver steatosis, while non-nutritive sweeteners generate gut dysbiosis that complicates the metabolic control exerted by the liver. The present work explores an alternative approach for sweetening through the use of a simple carob-pod-derived syrup. This sweetener consists of a balanced mixture of fructose (47%) and glucose (45%), as sweetening sugars, and a functional natural ingredient (D-Pinitol) at a concentration (3%) capable of producing active metabolic effects. The administration of this syrup to healthy volunteers (50 g of total carbohydrates) resulted in less persistent glucose excursions, a lower insulin response to the hyperglycemia produced by its ingestion, and an enhanced glucagon/insulin ratio, compared to that observed after the ingestion of 50 g of glucose. Daily administration of the syrup to Wistar rats for 10 days lowered fat depots in the liver, reduced liver glycogen, promoted fat oxidation, and was devoid of toxic effects. In addition, this repeated administration of the syrup improved glucose handling after a glucose (2 g/kg) load. Overall, this alternative functional sweetener retains the natural palatability of a glucose/fructose syrup while displaying beneficial metabolic effects that might serve to protect against the progression towards complicated obesity, especially the development of liver steatosis.

## 1. Introduction

One of the major challenges in the current fight against the epidemics of obesity and diabetes type 2 (diabesity) is the correct use of dietary sugars [1]. Additionally, the challenge stands not only in reverting the widespread excessive consumption of added sugars in processed foods, but also in achieving a correct use of sugars or non-caloric compounds as sweeteners [2,3]. As an example concerning added sugars, the use of soft drinks has increased in recent decades, paralleling the rise of metabolic diseases. The consumption of sugary drinks, even in young adults, ranges from 0.24 L/day in Europe to 0.45 L/day in the United States, with glucose or high-fructose corn syrup as the source of carbohydrate added [4,5]. However, beyond the excess of calories provided for these added sugars, the emergence of complicated obesity linked to this dietary pattern has opened the debate of how we can substitute added sugars to improve health [5,6]. Remarkably, the presence of high fructose in processed foods has been identified as an unnoticed source of additional metabolic harm, especially to the liver [7,8]. Intensive research in the last decade has revealed that the metabolic pathway of fructose affects, among others, uric acid production, de novo lipogenesis, gluconeogenesis pathways, intestinal barrier integrity, and the gut–liver axis. All these alterations favor the development of non-alcoholic fatty liver, obesity, dyslipidemia, and insulin resistance, and eventually enhance cardiovascular risk [7,8,9,10]. The severity of this situation has prompted the WHO to recommend a reduction in excessive fructose and glucose in the diet and the replacement of fructose with non-caloric substitutes as alternative sweeteners.

However, the use of these non-caloric sweeteners has also become a public health problem [1,2,3,4]. Their rapid widespread use has not been followed by a noticeable reduction in the diabesity problem. In fact, multiple studies have demonstrated that non-caloric sweeteners do not contribute either to an improvement in insulin resistance or to a reduction in body weight [11,12,13]. In addition, important safety issues have been raised, although most of them are surrounded by controversy. Thus, alternative non-caloric sweeteners have been blamed for affecting the intestinal microbiome and disrupting the intestinal barrier [14,15,16]; they have been found to produce alterations in the gut–brain axis, thus affecting appetite and insulin release modulation [3,17]; their consumption alongside pregnancy has been found to result in obesity and insulin resistance in the adult offspring [18,19]; and finally, they have been questioned because of their potential carcinogenic effects, although systematic reviews are inconclusive [20]. It is true that for each study demonstrating a negative problem derived from alternative sweeteners, there are studies claiming the opposite (for an extensive review see, for instance, [3]). However, after a careful analysis of the multiple preclinical and clinical studies published, we can conclude that these non-caloric sweeteners have a neutral effect on glycemic control and body weight homeostasis, they do produce the dysbiosis of gut microbiota, and they have potential transgenerational effects if taken alongside pregnancy (something proven in preclinical models, raising the need of further human research). From this perspective, they are not the solution for the diabesity epidemic [21].

Thus, the panorama on the use of sweeteners is shadowed by serious concerns on the use of high-fructose syrups as added sugars, and the lack of benefits derived from the use of non-caloric sweeteners. Additionally, this view is the one derived from taking the most conservative option: questioning the toxicity of these sweeteners, but placing a serious concern for their use in pregnancy and gut dysbiosis-related disorders. Thus, the challenge stands at the starting point: how do we control the use of added sugars while providing a safe, palatable, sweet flavor to foods? There are non-explored solutions that come from the observation that maximizing the negative contribution of certain natural nutrients is an incorrect strategy since these nutrients must be incorporated into the diet. Additionally, this strategy affects each macronutrient, from sugars to fats. As an example, the incorporation of saturated fatty acids into diets in obesity clinical trials has improved plasma lipid profiles, resulting in healthy outcomes [22] despite the extensive literature banning saturated fats. Moreover, we cannot forget that these natural nutrients have optimized detection systems in the body that elicit an adequate physiological response. As an example, only sugars, not non-caloric sweeteners, elicit an adequate gut–brain response to adapt behavior and metabolism [23,24]. Thus, we must reconsider the use of physiological mechanisms to achieve the goal of keeping a healthy metabolic balance while preserving palatable properties. Let us translate this idea to the sugar/sweetener dilemma: the use of a monosaccharide-equilibrated food as a sweetener can be healthy if we are able to help the body avoid the transformation of these sugars into fat, leading to bodyweight gain and insulin resistance. This can be achieved by using functional foods that incorporate both an equilibrated composition of glucose and fructose plus a functional natural ingredient capable of improving carbohydrate metabolism.

In the present study, we test this hypothesis by analyzing, in humans and rodents, the physiological impact of the consumption of a natural syrup prepared from the pods of carob trees (*Ceratonia siliqua*). This sweetener contains glucose, fructose and D-Pinitol. This inositol is an insulin sensitizer capable of keeping glycaemia while avoiding both unnecessary insulin secretion and the conversion of carbohydrates into fat depots [25,26,27,28].

## 2. Materials and Methods


**A. Human studies**


### 2.1. Carob Syrup

A standardized carob syrup (InnoSweet^®^) manufactured by Euronutra SL (Málaga, Spain) was used in the experimental approaches designed for the present study. Both human healthy volunteers and laboratory Wistar rats were used. To prepare the syrup, physical process and chromatographic separation technologies without chemical modification were used. Natural raw materials (carob pods from *Ceratonia siliqua*) were milled and mixed with hot water. After filtration and chromatographic separation, a natural syrup containing sugars (including inositols) was obtained and adjusted to 70 Brix degrees (see Table 1 for composition). Considering the sugar fraction, the composition of the lot used for the present experiments was: 45.6% glucose, 47.3% fructose, 0.5% sucrose, and 3.2% D-Pinitol.

### 2.2. Study Design

The main aim of the study was first to compare the impact of an oral administration of a 50 g carbohydrate dose (using either glucose, carob syrup or a commercial agave syrup) on plasma glucose concentrations to obtain the glycaemic index of carob syrup with respect to glucose. A secondary aim was to analyse glucose homeostasis by monitoring the plasma concentrations of the endocrine hormones regulating it: insulin, glucagon, ghrelin, circulating free fatty acids, β-hydroxybutirate, and ketone concentrations. To achieve these aims, we designed a study consisting of a randomised trial where subjects were randomly assigned to one of the following treatments: (A) 8 subjects received 100 mL of a water solution containing 50 g of glucose; (B) 9 subjects received 50 g of carbohydrates from a natural carob-pod-derived syrup (InnoSweet^®^) containing 45.6% glucose, 47.3% fructose, 0.5% sucrose and 3.2% D-Pinitol (50 g of syrup contains a dose of D-pinitol of 1600 mg or 22.8 mg/kg body weight in a person weighing 70 kg), diluted with drinking water to a final volume of 100 ml; and finally (C) 6 subjects received 50 g of carbohydrates from a commercial agave syrup sold as a table sweetener (Mercadona, Spain, containing 77.2 g of carbohydrates/100 g, of which more than 75% was fructose). A third aim was to monitor the impact of carob syrup on metabolically active pituitary hormones (prolactin (PRL), growth hormone (GH) and thyroid-stimulating hormone (TSH)). In addition, taking into consideration that D-Pinitol can be converted to D-Chiroinositol that is capable of modifying the function of the hypothalamus–hypophisis–gonadal axis, we also monitored gonadotrophins (luteinizing hormone (LH) and follicle-stimulating hormone (FSH)). Concerning the sample size, we calculated it using the Gpower program, version 3.1.9.2, Heirich Heine University, Düsseldorf, Germany). The main variable of the study was the area under the curve (AUC) derived of plasma glucose excursions after the oral intake of either glucose solution, carob syrup, or agave syrup. Considering that, compared with the standard glucose solution used (50 g), carob syrup will have around 25 g (50%) and agave syrup around 20 g (1/5) of glucose content, effect sizes of 1.6 and 2 for glucose were considered for carob syrup and agave syrup, respectively. That gave a sample size of 10 subjects per group for comparing glucose versus carob syrup, and a size sample of 7 for comparing glucose versus agave syrup. That means a total of 27 subjects. However, after the recruitment period was closed, we only recruited 23 subjects that were allocated to the 3 groups in a proportional distribution with respect to the calculated sample size.

### 2.3. Human Volunteers

Twenty-three healthy volunteers were recruited among healthy clinical and laboratory staff of the Laboratory of Neuropsychopharmacology of the Regional University Hospital of Málaga. The inclusion criteria for all subjects were (a) age range of 18–65 years, (b) the presence of a baseline capillary blood glucose < 5.6 nmol/L (100 mg/mL) measured with a glucose oxidase method after overnight fasting, (c) the absence of obesity (BMI > 30) and the absence of a diagnosed/treated metabolic disease. Subjects were interviewed for a present or past diagnosis of diabetes (glucose > 7.8 mmol/L after a standard glucose load), hypertriglyceridemia or hypercholesterolemia under treatment, as well as for a clinical record of past endocrine disorders, including thyroid gland dysfunction or treatment with glucocorticoids. Based on the inclusion criteria, exclusion criteria were pregnancy or lactation, fasting glycaemia > 5.6 mmol/L (100 mg/dL), fasting insulinemia determined by ELISA > 25 mIU/L, diabetes, or using a medication known to interfere with carbohydrate metabolism. All the studies were performed under fasting conditions (overnight fasting for 12 h). Any of the women included used contraceptive medication. The main data regarding the male and female participants can be found in Table 2.

### 2.4. Blood Sample Collection, Plasma Preparation and Glucose and Fructose Concentration Monitoring

After 12 h overnight fasting, a catheter was inserted into the antecubital vein of each subject, blood was extracted at baseline (minute 0; while still fasting) and immediately after (within 5 min) the oral intake of the assigned doses of either glucose, the natural carob syrup, or agave syrup. Then, blood samples were collected at 0, 15, 30, 45, 60, 90 and 120 min after intake. Individuals were allowed to eat 6 h after the last blood sample. All-time points were used for glucose and insulin concentrations. Time points in between 0′ (basal) and 120′ post intake were used for glucose homeostasis-related measures. Only 0′, 60′ and 120’ time points were used for the analysis of pituitary hormones. All the blood samples were collected in Vacutest tubes (Vacutest Kima S.r.l., Arzergrande, Italy, cat. number: #13560) and centrifuged at 2000× *g* for 10 min at 4 °C; the plasma was kept at −80 °C for further biochemical analysis. Right after the blood was extracted from the subjects at each time point, glucose blood concentrations were measured with a commercially available glucometer (AccuCheck, Roche, Germany). For fructose measurements, 0.15 mL of plasma was deproteinized by mixing with an equal volume of ice-cold 20% perchloric acid. Samples were vortexed and centrifuged for 15 min at 2500 rpm in a microfuge. Supernatants were injected with HPLC coupled to a refractive index detector for fructose monitoring using an external standard calibration curve for the calculation of the concentrations. The HPLC system consisted of a μBondapak/carbohydrate column, a solvent system of acetonitrile water (75:25), and a flowrate of 1.8 mL/min.

### 2.5. Quantitation of Pinitol: Liquid Chromatography–Mass Spectrometry Method

MS method: A quantitation method for D-Pinitol in human plasma using liquid chromatography coupled to mass spectrometry (LC-MS/MS) was validated. Analysis was carried out using an Agilent 1290 (Agilent Technologies, Santa Clara, CA, USA) liquid chromatographer and an Api4000 triple-quadrupole mass spectrometer (SCIEX). Quantitation was obtained using the multiple-reaction monitoring (MRM) mode of the transitions at m/z 195.2→109.0 (quantitative) and 195.2/80.0 (qualitative) for D-Pinitol, both with collision energy to 20 Ev and m/z 261.1→205.1 for the internal standard (IS) (see Figure 1A,B). IS was provided by MEDINA.

Mass spectrometry conditions consisted of: decluster potential (DP): 10 eV; GS1 and GS2: 45 psi; Tª: 600 °C; and ion spray: 5500 Ev. Chromatographic conditions were the following: the mobile phase (MP) consisted of 0.1% formic acid−[water:AcN] [90:10] (MP A) and 0.1% formic acid−[AcN:water] [90:10] (MP B). The gradient program was applied as follows: t = 0−0.5 min 95% MP B; t = 4.50–6.70 min 25% MP B; t = 6.80 95% MP B; t = 6.80–9:00 min 95% MP B. The flow rate was 0.3 mL/min and the run time was 9 min. The injection volume was 5 μL. The chromatographic column used was × bridge BEH amide (waters) with dimensions of 2.1 × 100 mm and a particle size of 3.5 μm. The oven column was maintained at 30 °C. The method performance was validated according to the FDA recommendations proposed in 2018 for selectivity, sensitivity, matrix effect, linearity, precision, accuracy and the recovery of plasma samples [29].

Sample preparation: An aliquot of 50.0 μL plasma was taken and 2% TFA plus 150 μL of cold methanol containing the internal standard was added. After vortex mixing for one minute, the samples were centrifuged for 15 min at 13,300 rpm. The temperature of the centrifuge was set at 4 °C. An aliquot of 160.0 μL of the supernatant was transferred to a vial for evaporation. The samples were reconstituted in 100 μL water/acetonitrile (80/20) and 0.1% ammonia (20%) for LCMS analysis.

### 2.6. Quantitation of Plasma Metabolites and Plasma Hormone Concentrations

Glucose homeostasis-related hormones: plasma levels of hormones regulating glucose homeostasis were determined by the human Enzyme-Linked Immunosorbent Assay (ELISA) method using commercial kits: insulin (EMD Millipore Corporation, Billerica, MA, USA, cat. number: #EZHI-14K), glucagon (Elabscience Biotechnology Inc, Wuhan, Hubei, China, cat. number: #E-EL-H2237), and active ghrelin (Kamiya Biomedical Company, Seattle, WA, USA, cat. number: #KT-364). All samples were assayed in duplicate within one assay, and results were expressed in terms of the particular standard metabolite.

For calculating glycaemic index, area under the curve (AUC) of % of plasma glucose variation over basal levels versus time after ingestion (Graph 1C) was obtained for each experimental subject receiving either glucose, carob syrup or agave syrup. The mean AUC for the glucose group was normalized to 100 and the glycaemic index is expressed of carob and agave syrups obtained as the % over the glucose AUC.

Free fatty acid plasma concentration was measured using a commercial kit (Sigma-Aldrich, Saint Louis, MO, USA, cat. number: MAK044) according to the manufacturer’s instructions. All samples were assayed in duplicate within one assay, and results are expressed in terms of the standards provided by the kit.

Insulin resistance was evaluated according to the Homeostatic Model Assessment for Insulin Resistance (HOMA-IR index), calculated using the following formula: HOMA-IR= ((fasting plasma insulin [uIU/mL] × fasting blood glucose [mmol/L])/22.5) [30].

The plasma levels of beta-hydroxybutyrate were determined using a commercial kit (Sigma Aldrich, Saint Louis, MO, USA, cat. number: MAK041) according to the manufacturer’s instructions; results are expressed in terms of the standards provided by the kit.

Plasma levels of β-Ketone were determined using a commercially available meter (GlucoMen^®^ areo 2K, A. Menarini Diagnostics, Badalona, Barcelona, Spain) and the corresponding test strips (GlucoMen^®^ areo β-Ketone Sensor, A. Menarini Diagnostics, Badalona, Barcelona, Spain) according to the manufacturer’s instructions. Results are expressed in mmol/L.

Plasma levels of pituitary hormones: follicle-stimulating hormone (FSH, expressed in mU/mL), human growth hormone-1 (GH, expressed in ng/mL), luteinizing hormone (LH, expressed in mU/mL), prolactin (PRL, expressed in ng/mL), and thyroid stimulating hormone (TSH, expressed in µU/mL) were determined by the multiplex immunoassay system using a commercial kit: Bio-Plex Pro™ RBM Human Hormone Panel 1 (Bio-Rad, Hercules, CA, USA, cat. number: #171AHR1CK). The plate was run on a Bio-Plex MAGPIX™ Multiplex Reader with Bio-Plex Manager™ MP Software (Luminex, Austin, TX, USA), as described in [31]. Hormone detection limits were: 0.11 mU/mL (FSH), 0.0076 ng/mL (GH), 0.061 mU/mL (LH), 0.022 ng/mL (PRL) and 0.012 µU/mL (TSH).


**B. Preclinical studies in rats**


### 2.7. Statement

Animal experimental procedures were carried out in accordance with the European Communities directive 2010/63/EU and Spanish legislation (Real Decreto 53/2013, BOE 34/11370–11421, 2013) and approved by the Bioethics Committee for Animal Experiments of the University of Malaga, Spain, in accordance with the ARRIVE guidelines [32]. Accordingly, all efforts were made to minimize animal suffering and to reduce the number of animals used.

### 2.8. Animals

The experiments were performed with 4-to-5-month-old male Wistar rats (Crl:WI(Han)) weighing 400 ± 20 g (Charles River Laboratories, Barcelona, Spain). The animals were kept under a standard condition (light regimen of 12/12 h, day/night) and under temperature and humidity control. The rats were fed on a standard pellet diet (STD) (3.02 Kcal/g with 30 Kcal% protein, 55 Kcal% carbohydrates and 15 Kcal% fat) and were purchased from Harlam (Tecklad, Madison, WI, USA). Water and food were available ad libitum. Animals were anaesthetized with intraperitoneal (ip) sodium pentobarbital (50 mg/Kg body weight) before being sacrificed by decapitation.

### 2.9. Drug Preparation and Experimental Design

Carob syrup (Innosweet)^®^ and pure D-Pinitol were generously provided by Euronutra SL. For acute oral administration experiments, they were dissolved in drinking water to an equivalent dose of 2 g glucose/kg body weight/rat for carob syrup, and 100 mg/kg/day of D-Pinitol. For subchronic administration in drinking water, we considered that a rat drinks 12 ml water/100 g body weight/day. Thus, we prepared a water solution containing 10 mg of D-Pinitol/12 ml of drinking water (37.2 g of syrup/L), that would allow us to administer 100 mg/kg of D-Pinitol daily. Acute glucose tolerance experiment: Either glucose (2 g/kg) or carob syrup (equivalent to 2 g/kg glucose) were administered by gavage to 12-hour-fasting rats. Plasma glucose was monitored with a commercially available glucometer (AccuCheck, Roche, Germany) at 0, 5, 10, 15, 30, 45, 60 and 120 min post-administration of the glucose/syrup solutions. Effects of the chronic ingestion of carob syrup: Two groups of male Wistar rats (*n* = 10) were used. One drank a carob syrup solution (adjusted to a daily dose of 100 mg/kg body weight of D-Pinitol), whereas the second drank tap water. After 8 days, the animals were tested for a glucose tolerance test: Rats were food-deprived for 18 h and given an intraperitoneal injection of 2 g D-glucose/Kg; blood samples were collected from the tail vein at 0 (basal level), 5, 10, 15, 30, 45, 60 and 120 min after injection, and blood glucose concentrations were measured with a commercially available glucometer (AccuCheck, Roche, Germany). Upon completion, the animals resumed drinking and were sacrificed on the morning of the 10th day, not being food-restricted the night before. A third group of rats received only 10 mgkg bw of D-Pinitol for drinking water. This group was designed to be a control for the potential D-Pinitol actions in the liver. They were also sacrificed after 10 days of D-Pinitol drinking.

### 2.10. Sample Collection

Immediately after animal sacrifice, blood, liver and brain samples were collected. Blood was centrifuged (2100 g for 8 min, 4 °C) and the plasma was kept at −80 °C for biochemical analysis. Liver and brain samples were flash-frozen in liquid nitrogen, then stored at −80 °C until analysis.

### 2.11. Measurement of Metabolites and Hormones in Plasma and Liver

The following plasma metabolites were measured using commercial kits according to the manufacturer’s instructions and a Hitachi 737 Automatic Analyser (Hitachi Ltd., Tokyo, Japan): glucose, uric acid, creatinine, bilirubin, triglycerides, and the hepatic enzymes glutamic oxaloacetic transaminase (GOT) and alanine aminotransferase (ALT). The plasma levels of glucose homeostasis-associated hormones were determined by the Enzyme-Linked Immunosorbent Assay (ELISA) method using commercial kits: leptin, insulin and ghrelin ELISA kits (EMD Millipore Corporation, Billerica, MA, USA, cat. number: #EZRADP-62K, #EZRMI-13K and #EZRGRT-91K, respectively). All plasma samples were assayed in duplicate within one assay, and results were expressed in terms of the particular standard hormone. In order to monitor the presence of oxidative stress, the lipid peroxidation was evaluated measuring thiobarbituric acid reactive substance (TBARS). Using a specific kit (Cell Biolabs Inc., San Diego, CA, USA, cat. number: STA-330) according to the manufacturer’s instructions, malondialdehyde (MDA) was measured, which is a split product of an endoperoxide of unsaturated fatty acids resulting from the oxidation of lipid substrates. Plasma levels of beta-hydroxybutyrate were determined using a commercial kit (Sigma Aldrich, Saint Louis, MO, USA, cat. Number: MAK041) and liver glycogen levels were measured by a commercial glycogen assay kit (Sigma-Aldrich, Saint Louis, MO, USA, cat. number: MAK016), both according to the manufacturer’s instructions, and results were expressed in terms of the standards provided by the kits.

Fat extraction from liver tissue was performed as previously described [33]. Briefly, according to the (modified) method of Bligh and Dyer [34], total lipids were extracted from frozen liver samples using chloroform–methanol (2:1, *v*/*v*) and butylated hydroxytoluene (0.025%, *w*/*v*). Then, two centrifugation steps (2800 g for 10 min, 4 °C) were carried out and the lower phase, containing lipids, was extracted. Each sample was dried using nitrogen and the liver fat content was calculated by subtracting the weight of the empty tube from the weight of the same tube containing the liver lipids. The data are expressed as a percentage of tissue weight.

### 2.12. Protein Extraction and Western Blot Analysis

Total protein from liver and hypothalamus (17 mg average weight) was extracted using ice-cold cell lysis buffer for 30 min, as previously described [35]. Fifty micrograms of protein were resolved on 4–12% (Bis-Tris) Criterion XT Precast Gels (Bio-Rad, Hercules, CA, USA, cat. number: 3450124) and electroblotted onto nitrocellulose membranes (Bio-Rad, Hercules, CA, USA, cat. number: 1620115). For specific protein detection the membrane was incubated overnight in TBS-T containing 2% BSA and the corresponding primary antibody. Antibodies against GS (#3886), phospho-GS (Ser641) (#3891), GSK-3β (#12456), phospho-GSK-3β (Ser9) (#5558), GSK-3α (#4337), phospho-GSK-3α (Ser21) (#9337), Akt (#9272), phospho-Akt (Ser473) (#9271), mTOR (#2972) and phospho-mTOR (Ser2448) (#2971) were purchased from Cell Signalling Technology Inc. (Danvers, MA, USA); Adaptin (#ab151720) from Abcam (Cambridge, UK); and phospho-GSK-3α/β (Tyr279/Tyr216) (#15648) from Merck Millipore (Burlington, MA, USA). Primary antibodies were detected using anti-rabbit or an anti-mouse HRP-conjugated secondary antibody as appropriate (Promega, Madison, WI, USA, cat. number: W4011 and W4021, respectively). Specific proteins were revealed using the ECL™ Prime Western Blotting System (GE Healthcare, Chicago, IL, USA, cat. number: RPN 2236), in accordance with the manufacturer’s instructions. Images were visualized in the ChemiDoc MP Imaging System (Bio-Rad, Hercules, CA, USA). After measuring phosphorylation proteins, the specific antibodies were removed from the membrane by incubation with a stripping buffer (2% SDS, 62.5 mM Tris HCL pH 6.8, 0.8% β-mercaptoethanol) for 30 min at 50 °C. Membranes were extensively washed in ultrapure water and then re-incubated with the corresponding antibody specific for total protein. The quantification of results was performed using ImageJ software (http://imagej.nih.gov/ij (accessed on 4 February 2020), Bethesda, MD, USA). The specific signal level for total proteins was normalized to the signal level of the corresponding Adaptin band of each sample in the same blot. The phosphorylation stage of a protein was expressed as the ratio of the signal obtained with the phospho-specific antibody relative to the appropriate total protein antibody. The amount of the protein of interest in the control samples was arbitrarily set as 1.

### 2.13. Real-Time qPCR

RNA isolation and cDNA synthesis:

Rat liver sections (50–80 mg) were homogenized on ice and total RNA was extracted from tissue using the Trizol^®^ method according to the manufacturer’s instructions (ThermoFisher Scientific, Waltham, MA, USA, EE.UU.). RNA samples were isolated with the RNAeasy MinElute cleanup-kit including digestion with the DNase I column (Qiagen, Hilden, Germany) according to the manufacturers’ instructions and quantified using a spectrophotometer Nanodrop TM ND-1000 (Thermo Fisher Scientific, Waltham, MA, USA, EE.UU.) to ensure A260/280 ratios of 1.8 to 2.0. Reverse transcription was carried out from 1 μg of mRNA in a reaction volume of 20 μL total using the qScript XLT cDNA SuperMix (Quantabio, Beverly, MA, USA).

Real-time qPCR and Gene Expression Analysis:

Real-time qPCR reactions were carried out in a CFX96TM Real-Time PCR Detection System (Bio-Rad, Hercules, CA, USA) for each cDNA template, and amplified in 10 µL reaction volume containing 4.5 μL of cDNA (previously diluted 1/100, that is, a total amount of 2.25 ng of cDNA per reaction) and 5.5 μL of PerfeCTa qPCR ToughMix (Quantabio, Beverly, MA, USA, EE.UU.) containing the corresponding primer. The gene-specific primers for the target rat genes: *Fbp1* (fructose 1,6 bisphosphatase 1), *G6pc* (glucose-6-phosphatase catalytic subunit), *Pc* (pyruvate carboxylase), *Pck1* (phosphoenolpyruvate carboxykinase 1), *Pklr* (pyruvate kinase liver/RBC), *Fasn* (fatty acid synthase), *Acox1* (acyl-CoA oxidase 1), *Acaca* (acetyl-CoA carboxylase alpha), *Cox4i1* (cytochrome c oxidase subunit IV isoform 1), *Cox4i2* (cytochrome c oxidase subunit IV isoform 2), *Scd1* (stearoyl-coenzyme A desaturase 1), *Cpt1a* (carnitine palmitoyltransferase 1a), *Actb* (beta actin), are shown in Appendix A. All primers were obtained based on TaqMan^®^ Gene Expression Assays and the FAM™ dye label format (ThermoFisher Scientific, Waltham, MA, USA, EE.UU.). Each reaction was run in duplicate. Cycling parameters were 50 °C for 2 min to deactivate single- and double-stranded DNA containing dUTPs, 95 °C for 10 min to activate Taq DNA polymerase followed by 44 cycles at 95 °C for 15 s for cDNA melting, and 60 °C for 1 min to allow for annealing and the extension of the primers, during which fluorescence was acquired. We found that a single product was amplified using a melting curve. For the relative quantification the mean of duplicates was used. The expression of *Actb* gene was unaffected during all experimental treatments. The *Actb* gene was chosen as reference gene and all results are normalized with respect to the water group.

### 2.14. Statistical Analysis

Graph-Pad Prism 8.0 software (GraphPad Software, Inc., San Diego, CA, USA) was used to analyze the data. Values are represented as mean ± standard error of the mean (SEM) for each experimental group, according to the assay. The significance of differences within and between groups was evaluated by a one-way or two-way (depending on the assay) analysis of variance (ANOVA) followed by post hoc test for multiple comparisons. In case data were not distributed normally, the Kruskal–Wallis rank test was performed, as indicated. A *p*-value ≤ 0.05 was considered to be statistically significant. (* = *p* < 0.05; ** = *p* < 0.01; *** = *p* < 0.001).

## 3. Results

### 3.1. Effects of Acute Administration of Carob Syrup on Plasma Levels of D-Pinitol in Humans

The oral administration of 50 gr of carob syrup, containing 1600 mg of D-Pinitol, resulted in a progressive increase in plasma D-Pinitol levels that peaked 60–90 min after the administration (F(5,54) = 3.24, *p* < 0.01) and remained stable for at least one hour. Detection was significant as early as 30 min after the ingestion, suggesting a rapid incorporation of D-Pinitol to the blood stream, and reflecting that the liver passage was not generating a total clearance of the compound once incorporated to the portal circulation (Figure 1A). We performed a correlation analysis of peak concentrations of D-Pinitol versus age and BMI and an analysis of gender differences. D-Pinitol correlates positively with age (r^2^ = 0.45, *p* (two-tailed) = 0.047). There were no correlations with D-Pinitol with BMI (r^2^ = 0.44, *p* (two-tailed) = 0.05). Mean peak plasma D-Pinitol values were similar in females (1527 ± 589 ng/mL) compared to male subjects (1562 ± 178 ng/mL).

### 3.2. Comparative Glucose Handling and Insulin and Glucagon Secretion Observed after the Ingestion of Glucose, Carob Syrup or an Agave Syrup Sweetener, and Calculation of the Glycemic Index of Carob Syrup

The oral administration of 50 g glucose generates a rapid excursion of plasma glucose levels that peaked at 30 min after administration, returning to basal levels only at the end of the 120 min test. The administration of 50 g of carbohydrates in the carob syrup also generated glucose excursions that were similar to peak values at 30 min after ingestion, but recovered much more rapidly, returning to normality within the first hour after ingestion. Agave syrup intake produced a lower increase in glycaemia, but this was sustained for the 120 min of the study. A comparative analysis of the area under the curve (AUC) revealed that individuals taking glucose stayed under hyperglycaemia for longer compared to those receiving either carob syrup or agave syrup (F(2,19) = 9.6, *p* < 0.002). Interestingly, despite the peak of glucose observed at 30 min, the AUC of agave and carob syrup was found to be equal, because of the rapid return to basal levels observed in the carob syrup group (Figure 1B). This was confirmed by analyzing the % of change over basal glucose values, a better index for the physiological response to glucose excursions. Again, AUC revealed that subjects receiving glucose had greater long-lasting variations from basal levels of glucose (F(2,19) = 6.2, *p* < 0.009), with carob syrup being the most efficient sweetener to return to basal glucose levels (Figure 1C). We used the AUC of glucose normalized to 100 to calculate the glycemic index of carob and agave syrups. Carob syrup had a glycemic index of 73.4 ± 6.9, whereas agave syrup had a glycemic index of 76.2 ± 3.3. These results indicate that despite having a double amount of glucose compared to agave syrup, the presence of D-pinitol helps to rapidly retire the excess of glucose, resulting in an equal glycemic index. As expected, the glucose excursions caused a rapid insulin response in subjects receiving glucose or carob syrup, but not fructose-based agave sweetener (Figure 1D). AUC analysis again revealed that the insulin response was shorter and less intense in carob syrup-receiving subjects (F(2,19) = 47.4, *p* < 0.001). Concerning plasma fructose, as expected, agave syrup generated a greater and more prolonged rise in plasma fructose levels than carob syrup, which immediately reduced the rise in fructose observed 15 and 10 min after the intake of the syrup (F(1,73) = 29.06, *p* < 0.0001). Area under the curve was clearly different when carob syrup and agave AUC were compared (AUC carob syrup 1471 ± 232; AUC agave 2534 ± 249, t = 8.44, df = 13, *p* < 0.001) (Figure 1E). Considering the glucagon response, the three groups had a similar profile (Figure 1F). However, the glucagon/insulin ratio was higher in the carob syrup group with respect to the glucose group (time × treatment interaction (F(4,60) = 4.16, *p* < 0.005), revealing a lesser need for insulin and the more active role of glucagon in the carob-syrup-treated individuals (Appendix A).

### 3.3. Effects of the Ingestion of Glucose and Carob Syrup on the Plasma Concentrations of Ghrelin, Free Fatty Acids, β-Hydroxybutirate and Pituitary Hormones

Plasma ghrelin (Figure 1G) concentration was not affected by either glucose or carob syrup. Free fatty acid plasma concentration (Figure 1H) decreased in subjects receiving glucose, an effect that appeared 45 min after the ingestion of this monosaccharide, and lasted up to 120 min (F(6,28) = 8.98 *p* < 0.001)). This effect was not observed in the individuals receiving carob syrup, where the free fatty acid concentrations remained stable (F(6,31) = 0.28 *p* = 0.94, non-significant). Plasma concentrations of β-hydroxybutirate (Appendix A) were found to decrease with time equally in both glucose- and carob-syrup-treated individuals (F(2,42) = 10.8 *p* < 0.001). However, there were no differences in between both treatments (F(1,42) = 1.7 *p* = 0.2, non-significant). Subjects did not exhibit elevated levels of plasma ketones despite being fasting, and neither glucose nor carob syrup enhanced the concentration of ketones (Appendix A). Finally, none of the pituitary hormones analyzed (prolactin, thyroid-stimulating hormone, and the gonadotropins LH and FSH) were affected by the treatment with carob syrup (Appendix A). A sex difference was observed for LH (F(1,21) = 4.62, *p* < 0.05) and FSH (F(1,21) = 4.94, *p* < 0.05).

### 3.4. Effects of the Acute and Repeated Administration of Carob Syrup on Glucose Handling in Rats

In order to further explore the functional properties of the carob pod syrup, we tested its effects of glucose handling in male Wistar rats. First, we tested if the presence of D-Pinitol affected glucose excursions in a differential way to that observed for glucose alone. Figure 2A shows that the administration of carob syrup at a dose containing 2 g/kg glucose (plus 2.1 g/kg of fructose and 130 mg/kg D-Pinitol) resulted in less intense hyperglycemia (F(1,144) = 69.1 *p* < 0.001). Area under the curve revealed that the time spent under hyperglycemia was less than that spent by the animals receiving the same amount of glucose without D-Pinitol (t = 7.81, df = 18, *p* < 0.001). These results are similar to those observed in humans. Next, we tested if the repeated administration of the carob syrup (at a dose of 100mg/kg/day of D-Pinitol for 10 days) was sufficient to improve glucose handling, after an oral load with 2 g/kg glucose. Figure 2B demonstrates that the intake of this carob syrup reduced the intensity of hyperglycemia when compared with animals fed with the standard chow (F(1,144) = 12.1 *p* < 0.001)). Again, area under the curve analysis showed that the time spent under hyperglycemia in animals fed with the carob syrup was less than the time spent by animals fed with standard chow and water (t = 3.35, df = 18, *p* < 0.003).

### 3.5. Metabolic Effects of Repeated Administration of a Carob Syrup in Rats

As described above, excessive glucose consumption has been thought to increase de novo lipogenesis, resulting in liver fat depots while promoting uric acid production and boosting gluconeogenesis, favoring hyperglycemia. In order to control if the consumption of carob syrup results in alterations in both plasma and liver biochemistry, several analytical controls were performed and displayed in Table 3. Carob syrup consumption did not alter the major indexes of liver or kidney function (normal levels of AST, uric acid, bilirubin, creatinine or the oxidative stress indicative compound malonyl dialdehyde). Only plasma urea concentration was increased (t = 2.98, df = 18, *p* < 0.01), suggesting enhanced urea production by the liver probably derived of the described actions of D-Pinitol as a promoter of glucagon effect, and not derived from kidney functioning, since creatinine clearance was preserved. Neither insulin, glucagon nor leptin hormones involved in glucose and appetite homeostasis were affected by carob pod consumption. However, ghrelin concentrations, a hormone found to be increased after the acute administration of D-Pinitol, were higher in animals receiving carob syrup through drinking water (t = 3.43, df = 18, *p* < 0.01). The liver content of fat was decreased after carob syrup consumption (t = 9.6, df = 18, *p* < 0.001). This effect was not linked either to the export of triglycerides or to the production of b-hidroxybutyrate, which were found not to be affected by carob syrup consumption. Finally, liver glycogen contents were found to be reduced in animals consuming carob syrup) (t = 5.65, df = 18, *p* < 0.01). The effects on urea production and liver fat content were replicated when a dose of 100 mg/kg of D-Pinitol alone was given to the animals through drinking water for 10 days (Appendix A).

### 3.6. Effects of Repeated Administration of a Carob Syrup on the Liver Expression of Key Enzymes for Neoglucogenic and Lipid Metabolism Pathways in Rat Liver

Because of the reduction in liver content for both fat and glycogen, we examined by real-time PCR the expression of the key enzymes of gluconeogenesis, de novo lipid synthesis, and fatty acid oxidation pathways. Figure 3 illustrates how carob syrup consumption increases the expression of pyruvate kinase (t = 2.2 df = 18, *p* < 0.05), phosphoenolpyruvate carboxykinase (t = 3.39, df = 18, *p* < 0.01), and the catalytic subunit of the glucose-6 phosphatase (t = 2.77, df = 18, *p* < 0.05), suggestive of an increased activity of neoglucogenic pathways. Figure 4 depicts that the gene expression of any of the enzymes promoting the de novo synthesis of lipids were affected by the consumption of carob syrup. However, a clear increase in the gene expression of relevant enzymes related to fatty acid oxidation was detected after the administration of this natural sweetener. These enzymes were acyl-coenzymeA oxydase (Mann–Whitney U = 21.5, *p* < 0.03), carnitine palmitoyltransferse 1A (t = 2.2, df = 18, *p* < 0.05) and cytochrome C oxydase isoform 4 (Mann–Whitney U = 20, *p* < 0.05).

### 3.7. Effects of Repeated Administration of Carob Syrup on Insulin Signaling in the Liver, the Hippocampus, and the Hypothalamus of Rats

Since previous works [25] have described that D-pinitol alters the glucagon/insulin ration, promoting an activation of the gluconeogenesis, we explored the status of insulin signaling in the liver and the brain by Western blot analysis. Figure 5 shows that carob pos syrup enhanced the inhibitory phosphorylation of glycogen synthase kinase 3 and isoforms α (t = 2.49, df = 8, *p* < 0.05) and β (t = 2.33, df = 13, *p* < 0.05) without affecting the phosphorylation status of protein kinase B/AKT, glycogen synthase, or the mammalian target of rapamycin (mTOR). Thus, we did not observe a recruitment of the glycogen synthesis pathway at liver tissue. In the hypothalamus (Appendix A) we found an activatory phosphorylation of glycogen synthase (t = 3.3, df = 10, *p* < 0.01) and mTOR (t = 3.1, df = 10, *p* < 0.01), and a tendency to display a greater phosphorylation of AKT (t = 2.05, df = 10, *p* = 0.06). In the hippocampus (Figure 6), we found no effects of carob syrup on the insulin signaling chain, with the exception of a decreased activatory phosphorylation of GSK3β (t = 2.6, df = 184, *p* < 0.03) and a clear decrease in the phosphorylation of the microtubule-associated protein tau (t = 3.2, df = 13, *p* < 0.01), a specific pharmacological target of D-Pinitol that we have recently described to be mediated by cyclin-dependent kinase 5 (CDK5).

## 4. Discussion

The rationale for the present experiment is to provide evidence of the feasibility of improving the normal physiological response to caloric sweet intake by boosting endogenous modulatory mechanisms that help both to keep weight and to sustain energy expenditure. Additionally, we aimed to do so by using a simple design: a balanced composition of fructose and glucose and a natural insulin sensitizer capable of boosting glucagon/ghrelin secretion while lowering insulin requirements: D-Pinitol [25,26,27,36,37,38,39]. A major criticism for the use of complex extracts stands precisely on the unknown composition that makes it impossible to fully understand the pharmacological properties of these extracts. The use of a simple composition such as glucose/fructose/D-pinitol allows us to better understand both pharmacological effects and physiological impact.

Our study in human volunteers reflects clearly that a natural syrup containing only glucose, fructose and D-Pinitol, keeping sweet palatability and the corresponding caloric value, produces glucose excursions that are shorter in time than those observed for glucose intake alone. Moreover, both the percentage of change over basal glucose levels and the insulin release induced by the intake of the carob syrup had less intense and shorter durations than those observed for glucose. It is true that the fructose-based sweetener (agave syrup) had better glucose and insulin-associated responses, but as we have described, fructose has a dark metabolic side that we should avoid [7,8,9,10], and thus its reduction is recommended (again, not its complete suppression, unless there is a relevant medical intolerance). The results obtained in humans suggest that by adding D-Pinitol we can modify the glycemic index of glucose/fructose syrup, improving glucose handling and insulin responses. Interestingly, the presence of fructose and glucose in the syrup is not associated with a decrease in circulating free fatty acids, something described for both glucose (see the decrease in circulating FFAs in Figure 1H) and especially fructose [40], suggesting that D-pinitol prevents the antilipolytic-prolipogenic actions of fructose. Nonetheless, plasma fructose concentrations after the intake of carob syrup are time-limited, as is the case with glucose, suggesting a relevant role of D-Pinitol on its regulation. Compared with agave syrup, the duration and intensity of the plasma peak of fructose was much lower. However, the presence of fructose demands more studies to clarify the potential unwanted side effects derived of the repeated consumption of carob syrup. The additional preclinical studies designed have addressed most of these potential problems, as well as added more information regarding the recently described properties of D-Pinitol [25,41,42].

First, we wanted to confirm that the lowering of glucose excursion observed after the administration of the carob syrup in humans was not derived of the different amount of glucose administered. We used Wistar rats to analyze the glucose handling after the administration of glucose (2 g/kg) or an amount of syrup with the same dose of glucose (2 g/kg glucose plus 2.1 g/kg fructose). The results clearly indicate that despite the presence of both glucose and fructose, glycaemia variation was substantially lower in animals receiving the syrup, supporting the insulin sensitizer activity of D-Pinitol. Moreover, the administration of carob syrup at a dose of 100 mg/kg daily (equivalent to supplementation with 1.48 g/kg day of fructose and 1.43 g/kg/day of glucose) clearly improved the glucose handling when an oral glucose load of 2 g/kg was given to the animals. Although, as we will discuss below, experimental data suggest that this effect derives both direct actions of D-pinitol and metabolic adaptions induced by this inositol, and thus we cannot exclude that modifications in metabolically active tissues such as muscle and adipose tissue might contribute to the improvement in glucose handling. We did not measure food intake or energy expenditure in this animal model, although preliminary data (Appendix A) suggest that D-Pinitol does not affect feeding behavior.

This modulation of carob syrup could be observed in other metabolic parameters, and was partially replicated in a parallel experiment using only D-Pinitol at the same dose (Appendix A). Thus, after 10 days of ingestion of the syrup, animals had enhanced ghrelin secretion, and a lowering of liver depots of fat associated with a reduction in glycogen. A potential explanation for the decrease in the liver content of fat is that carob syrup exposure redirects liver metabolism towards glucose export and lipid oxidation. We attribute these effects to D-Pinitol, since previous studies with this inositol [25] have produced the same set of effects. In this study, we observed a decrease in insulin secretion associated with a rise in ghrelin and glucagon. Since D-Pinitol is able to activate both glucose uptake in muscle cells [27] and canonical insulin signaling in the hypothalamus [41], our interpretation is that D-Pinitol favors glucose transfer from liver to other tissues, including muscle. We do not know the impact of D-Pinitol (given alone or under the carob syrup format) on either adipose tissue function or energy expenditure, so future studies must address these important aspects to understand the pharmacological profile of this inositol. In addition, plasma concentrations of urea, but not uric acid, were also increased, suggesting that the glucagon pathway was activated. Glucagon is a hormone secreted by the α-cells of the pancreatic islets that oppose insulin and sustain glycaemia by promoting gluconeogenesis and releasing glucose from liver glycogen deports [43]. In addition, glucagon inhibits the de novo synthesis of lipids, boosting the use of amino acids and enhancing urea production [44,45]. These physiological actions are coincident with the metabolic profile we found in animals receiving the carob syrup, where we found a decrease in both fat and glycogen content in the liver, and an increase in plasma urea. D-Pinitol has been described to enhance the glucagon/insulin ratio [25], a finding we have replicated in humans (see Appendix A), although the data obtained in the preclinical study in rats in the present report were not significant (Table 3). Probably, the effects D-pinitol on glucagon secretion are associated with the syrup drinking pattern, so we did not observe it at the moment of sacrifice of the animals. Nonetheless, we did observe a potentiation of the expression of the gluconeogenic pathway (an enhancement of some of the main enzymes for the de novo synthesis of glucose, including pyruvate kinase, phosphoenolpyruvate carboxykinase, and the catalytic subunit of glucose-6 phosphatase), and an enhanced expression of the fatty acid oxidation pathway (acyl-CoenzymeA oxydase, carnitine palmitoyltransferse 1A and cytochrome C oxydase isoform 4). These effects on lipid oxidation were not coupled to the plasma export of b-hydroxybutyrate, nor resulted in ketone production, indicating that fatty acid oxidation was not used to generate ketone bodies as glucose suppliers. In addition, we did not find evidence for oxidative stress, since malonyl dialdehyde in the liver was not enhanced after carob syrup consumption. Thus, the overall profile of carob syrup is that of a safe sweetener in the short term; it is capable of precluding the unwanted effects of fructose, and favors the actions of glucagon, which are seriously compromised in liver steatosis. Further studies using long-term exposure (4 to 8 months) to carob syrup are necessary to fully ensure the lack of fructose-associated toxicity derived of its use.

The present results are in accordance with the described actions of D-Pinitol, that eventually prompted clinical trials for positioning this inositol as a food supplement for the treatment of non-alcoholic steatohepatitis [28]. In addition, the effects observed on the insulin signaling cascade in the liver, after 10 days of consumption of the carob syrup, suggest an inhibitory phosphorylation of GSK3α and GSK3β, thus precluding the activation of glycogen synthase, which was not affected by the treatment. This finding suggests that the syrup does not promote glycogen storage, being in accordance with the reduced liver glycogen found. The present study cannot explain whether the effects observed are solely derived from the presence of D-Pinitol, or if they come from the association of this inositol to glucose and fructose. In a parallel experiment, we found increased urea production and a decrease in the contents of fat in the liver after exposure to D-Pinitol alone for 10 days. In addition, the profile of the effects of the syrup on the insulin signaling cascade found in the hippocampus and the hypothalamus were very similar to those described in experimental conditions where D-Pinitol was administered alone [41,42]. In the hippocampus, the acute and repeated administration of D-Pinitol did not affect insulin signaling, but resulted in the dephosphorylation of tau protein. This is exactly the same profile found in the hippocampus of the animals fed on the carob syrup [42]. The possibility of the neuroprotective actions of this syrup composition in tauopathies will have to be studied to fully confirm this possibility, opening an additional utility for inositols in aging and neurodegeneration [46]. We have observed that the effect on tau phosphorylation could be replicated also in a 3TG transgenic model of Alzheimer’s disease. In the case of the hypothalamus, where D-Pinitol administration produces a general activation of AKT signaling cascade, the effects observed in the animals fed with the syrup demonstrated a similar pattern to the phosphorylation of GSK3β, the mammalian target of rapamicine (mTOR), and glycogen synthase [41]. Taken all together, the results suggest that D-Pinitol present in the syrup is bioactive and that the presence of glucose and fructose does not affect its pharmacological properties.

### Limitations of the Present Study

The present study explores the acute actions of a carob syrup in a reduced number of human volunteers. Future studies using a within-subject design (the same subject receiving all the treatments) and a higher number of male and female volunteers are needed to better compare the actions of carob syrup with respect to other sweeteners such as agave or high-fructose corn syrup (HFCS). In addition, there is a need for studying the impact of this type of functional sweetener in the diabetic population. The toxicity derived from very long-term exposure to carob syrup in preclinical models must be addressed, especially in comparison with the well-known toxicity derived from chronic exposure to high-fructose-containing sweeteners such as agave or HFCS.

## 5. Conclusions

The use of natural syrup formulae containing glucose, fructose and D-Pinitol had a safe profile and resulted in a better glycemic index than that observed when glucose was given alone. This was likely caused by the reduction in the insulin response to increased glycaemia and the promotion of glucagon activity. The repeated administration of this syrup to male rats did not produce lipid depots, oxidative stress, or liver/kidney toxicity, as has been reported to occur with fructose when given alone. Overall, the present study is a positive proof of concept of the use of formulae capable of retaining sweetening properties, caloric value, and a safe metabolic profile. Whether the use of this “functional sweetener” is useful in the context of obesity and diabetes will require additional research.

## Figures and Tables

**Figure 1 pharmaceutics-14-01594-f001:**
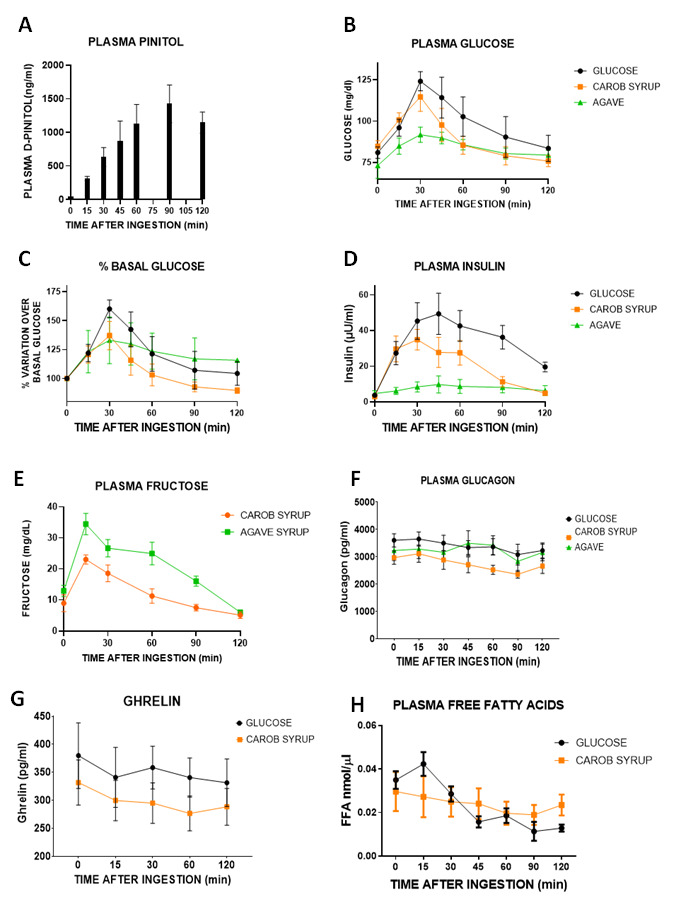
Acute effects of (1) a single oral dose of a glucose solution (50 g in 100 mL of water), (2) a natural carob-pod-derived syrup (Innosweet^®^, 50 g of carbohydrates in 100 mL water containing equal amounts of glucose and fructose and 1600 mg of D-Pinitol) or 50 g of carbohydrates in 100 mL of water from a commercial agave syrup (containing > 37.5 g of fructose). (**A**) Time-course of D-Pinitol plasma levels after carob syrup ingestion; (**B**) evolution of glucose concentrations after the ingestion; (**C**) percentage of variation of plasma glucose over basal levels previous to ingestion; (**D**) plasma insulin concentrations; (**E**) Insulin Resistance Index (homeostatic model) at different points after the ingestion; (**F**,**G**) evolution of plasma concentration of glucagon (**F**), total ghrelin (**G**) and free fatty acids. (**H**) Data are means ± standard error of the mean of 8 subjects for glucose, 9 subjects for carob syrup, and 6 subjects for agave sweetener.

**Figure 2 pharmaceutics-14-01594-f002:**
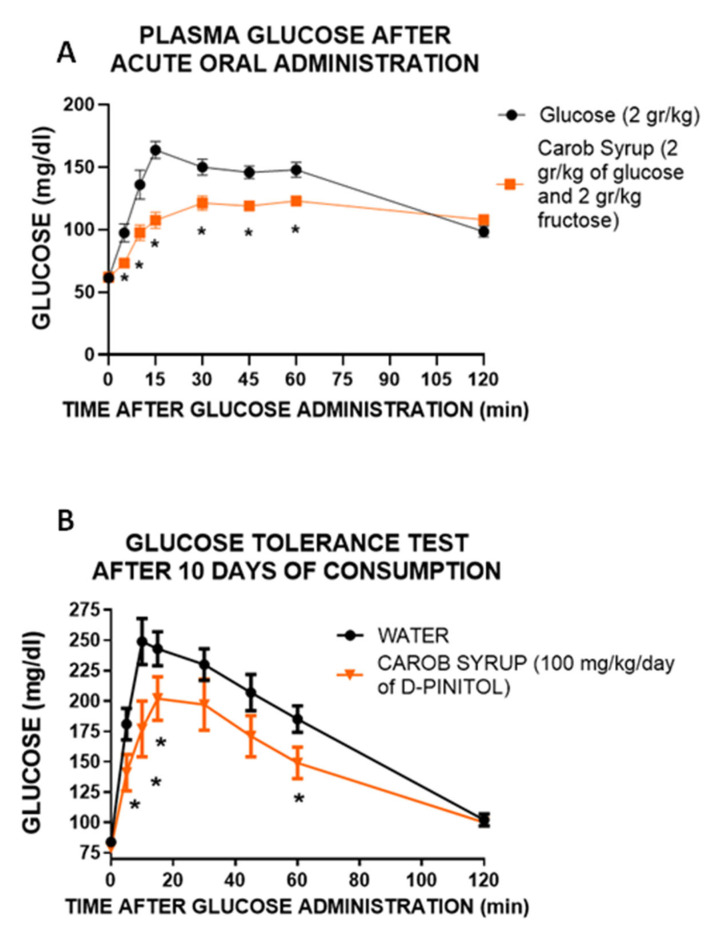
(**A**) Acute effects of (1) a single oral dose of a glucose solution (2 g/kg) or (2) a single oral dose of a carob syrup containing glucose (2 g/kg), fructose (2.1 g/kg) or D-Pinitol (130 mg/kg), on plasma glucose levels in adult male Wistar rats, measured 120 min after administration. (**B**) Time-curse of plasma glucose levels after an oral glucose load (2 g/kg) in both control animals and animals drinking water mixed with carob syrup (up to a 100 mg/kg D-Pinitol per day of study) for 10 days. In both cases, the glycaemic response was better in animals receiving the carob syrup. Data are means ± standard error of the mean for at least 8 subjects per treatment group (* *p* < 0.05 carob syrup vs. control (water) group).

**Figure 3 pharmaceutics-14-01594-f003:**
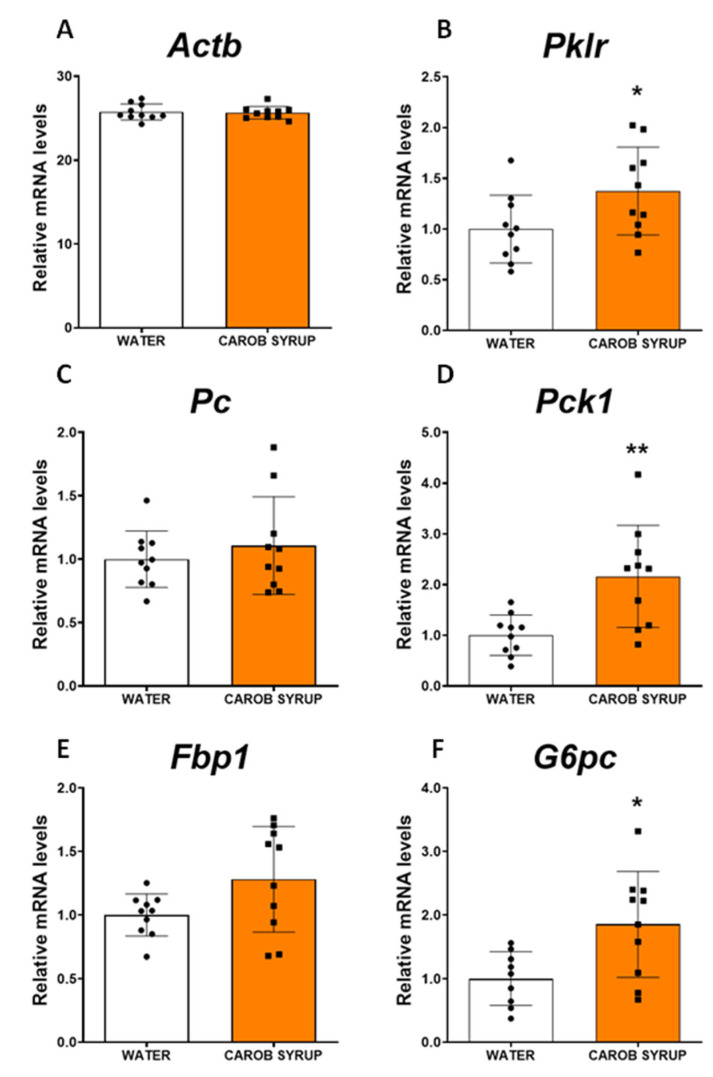
Effects on the liver neoglucogenic pathway of carob syrup mixed in drinking water at a dose of 100 mg/kg D-Pinitol per day of study with repeated administration for 10 days. Panels depict the quantitative expression of mRNA coding for (**A**) actin B (Actb, housekeeping gene), (**B**) pyruvate kinase (Pklr), (**C**) pyruvate carboxylase (PC), (**D**) phosphoenolpyruvate carboxykinase (Pck1) (**E**) fructose bis phosphatase (Fbp1), and (**F**) the catalytic subunit of glucose-6 phosphatase (G6pc). Data are means ± standard errors of the mean of at least 8 determinations per group. (*) *p* < 0.05, (**) *p* < 0.01 carob syrup vs. water drinking control animals.

**Figure 4 pharmaceutics-14-01594-f004:**
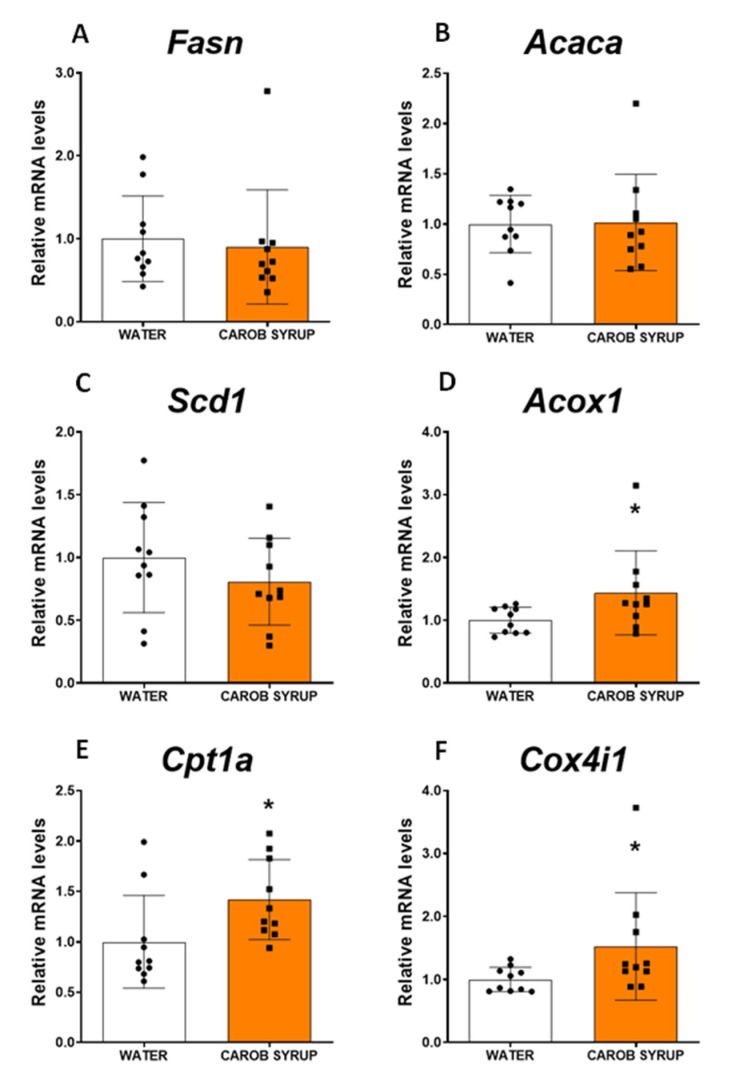
Effects of repeated administration of carob syrup for 10 days on liver lipid synthesis and oxidation pathways. Panels depict the quantitative expression of the mRNA coding for (**A**) fatty acid synthase (Fasn), (**B**) acetyl-coenzyme-A carboxylase (Acaca), (**C**) stearoyl-coenzymeA desaturase-1 (PC), (**D**) acyl-coenzyme-A oxidase-1 (Pck1), (**E**) carnitine palmitoyltransferase 1 (Cpt1), and (**F**) cytochrome C oxydase 4 isoform 1 (Cox4i1). Data are means ± standard errors of the mean of at least 8 determinations per group. (*) *p* < 0.05 carob syrup vs. water drinking control animals.

**Figure 5 pharmaceutics-14-01594-f005:**
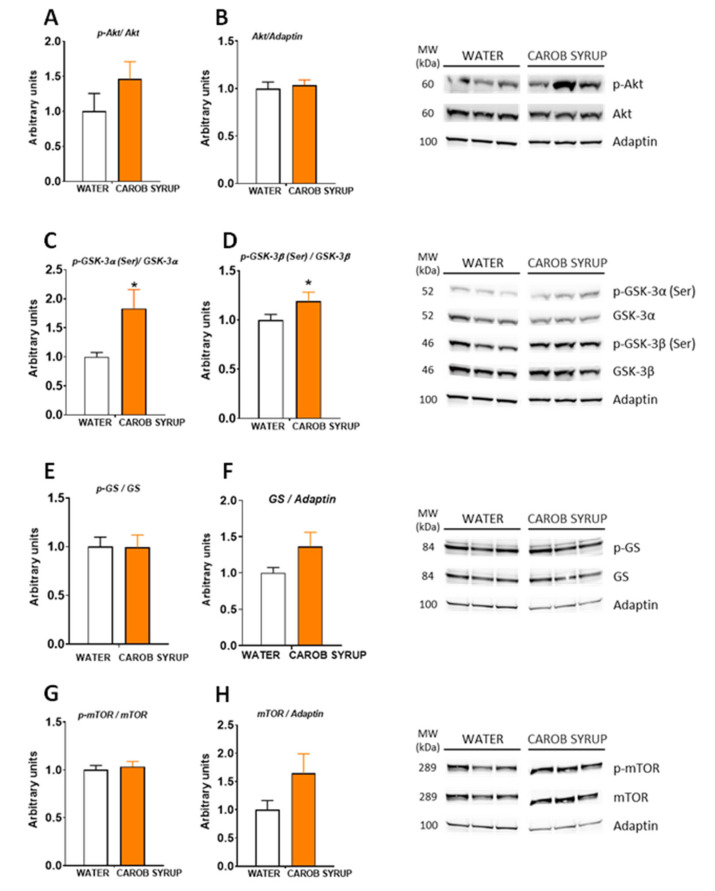
Effects of repeated administration of carob syrup on liver insulin signaling cascade for 10 days measured by Western blot analysis. (**A**) Phospho-protein kinase b/AKT (p-AKT), (**B**) protein kinase b/AKT (AKT), (**C**) phospho-glycogen synthase kinase 3 α (p-GSK3 α), (**D**) phospho-glycogen synthase kinase 3 β (p-GSK3 β), (**E**) phospho-glycogen synthase (pGS), (**F**) glycogen synthase (GS), (**G**) phospho-mammalian target of rapamicin (p-mTOR), and (**H**) mammalian target of rapamicin (mTOR). Data are means or adaptin-normalized band densities ± standard errors of the mean of 5–8 determinations per group. (*) *p* < 0.05 carob syrup vs. water drinking control animals.

**Figure 6 pharmaceutics-14-01594-f006:**
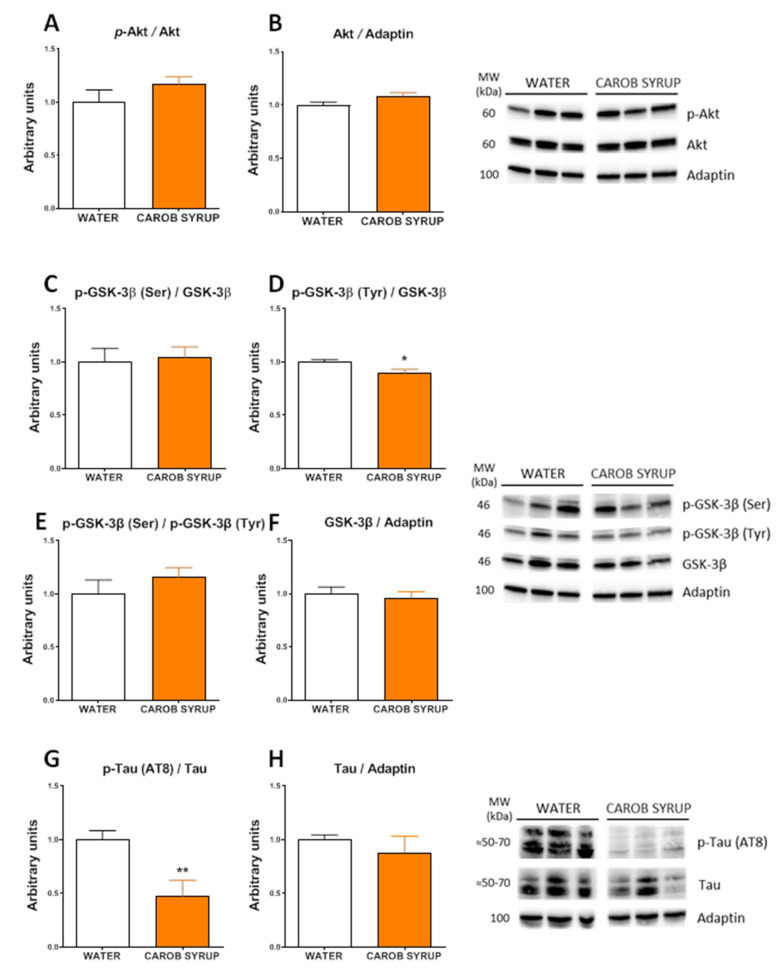
Effects of repeated administration of carob syrup on hippocampus AKT-signaling cascade and tau phosphorylation for 10 days, measured by Western blot analysis. (**A**) Phospho-protein kinase b/AKT (p-AKT), (**B**) protein kinase b/AKT (AKT), (**C**) serine-phosphorylated glycogen synthase kinase 3 β (p-GSK3 β)(Ser), (**D**) tyrosine-phosphorylated glycogen synthase kinase 3 β (p-GSK3 β)(Tyr)), (**E**) ratio p-GSK3 β)(Ser)/p-GSK3 β)(Tyr), (**F**) total GSK3 β, (**G**) phospho-tau (Ser202, Thr205) marked with AT8 antibody (p-TAU(AT8)) and (**H**) total tau. Representative westerns of each target are represented on the right panels. Data are means or adaptin-normalized band densities ± standard errors of the mean of 5–8 determinations per group. (*) *p* < 0.05, (**) *p* < 0.01 carob syrup vs. water drinking control animals.

**Table 1 pharmaceutics-14-01594-t001:** Composition of the carob syrup (Innosweet) used in the study.

	Value	Comments
BRIX at 20 °C	69–71	ISO1743
DENSITY at 20 °C	1.35–1.36	g/cm^3^
GLUCOSE	40	% dried matter
FRUCTOSE	45	% dried matter
SUCROSE	<5	% dried matter
D-PINITOL	>3	% dried matter
NON-SUGARS	<2.5	% dried matter

**Table 2 pharmaceutics-14-01594-t002:** Characteristics of human healthy volunteers participating in the present study.

	Males	Females
N	13	10
AGE (years)	37.7 + 12.2	34.3 + 11.6
WEIGHT (kg)	84.6 + 9.5	66.6 + 12.4
BODY MASS INDEX	27.2 + 3.7	23.6 + 4.0
BASAL GLUCOSE (mg/dl)	81.8 + 14.8	78.8 + 8.4
BASAL INSULIN (mIU/L)	3.5 + 1.4	2.95 + 1.8
HOMA-IR	0.66 + 0.28	0.58 + 0.36

Data are means + standard deviation.

**Table 3 pharmaceutics-14-01594-t003:** Plasma and liver biochemistry parameters after 10 days of drinking **water** or water-diluted carob syrup (equivalent to 100 mg/kg b.w./day of D-Pinitol).

	Water	Carob Syrup
N	10	10
Glucose (mg/dL)	247.0 ± 63.8	269.4 ± 46.3
CCreatinin (mg/dL)	0.57 ± 0.38	0.73 ± 0.15
Urea (mg/dL)	21.6 ± 3.2	**40.7** **± 5.9 (*)**
Bilirubin (mg/dL)	0.10 ± 0.09	0.13 ± 0.08
Uric Acid (mg/dL)	1.67 ± 0.29	1.90 ± 0.30
Triglycerides (mg/dL)	146.9 ± 30.1	156.3 ± 31.24
β-Hydroxy butirate (mg/dL)	1005 ± 86	1000 ± 74
AST (U/L)	152.6 ± 46.5	195.3 ± 97.4
Insulin (ng/mL))	14.9 ± 1.9	15.0 ± 1.6
Glucagon/Insulin ratio	26.6 ± 4.9	35.6 ± 9.4
Leptin (ng/mL)	14.8 ± 4.6	15.3 ± 4.3
Ghrelin (ng/mL)	0.42 ± 0.13	**0.67** **± 0.19 (*)**
TBARS (Malonyl dialdehyde, μM)	11.1 ± 2.7	11.6 ± 3.8
Total Fat in Liver (mg/g)	40.8 ± 1.3	**34.8** **±1.5 (*)**
Liver Glycogen (μg/g)	137.7 ± 34.1	**76.1** **± 5.2 (*)**

Data are means ± standard deviation. (*) indicates *p* < 0.05, ANOVA or Kruskal–Wallis test. AST (aspartate aminotransferase), TBARS (tiobarbituric acid-reactive species).

## Data Availability

All data generated or analysed during this study are available upon request by email to fernando.rodriguez@ibima.eu as a raw data file.

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
