# Peer review of "Endocrine and Metabolic Impact of Oral Ingestion of a Carob-Pod-Derived Natural-Syrup-Containing D-Pinitol: Potential Use as a Novel Sweetener in Diabetes"

_pharmaceutics, 2022, doi:10.3390/pharmaceutics14081594_

Round 1

Reviewer 1 Report

This manuscript presents the exploration of an alternative approach for sweetening using a simplified carob pod-derived syrup consisting of a balanced mixture of fructose, glucose, and a functional natural ingredient (D-Pinitol). The administration of this syrup to healthy volunteers resulted in less persistent glucose excursions, lower insulin response to the hyperglycemia produced by its ingestion and enhanced glucagon/insulin ratio, than that observed after the ingestion of glucose. Daily administration of the syrup to Wistar rats lowered fat depots in the liver, reduced liver glycogen, promoted fat oxidation, and improved glucose handling.

A couple of questions for the authors.

1.       On page 5, line 170-173. The chromatography gradient does not look right to me. Please check carefully. It should start with high percentage of MP A and gradually increase the percentage of MP B till 95% for 6.80-9:00 min. But from the manuscript, it started with 95% of MP B from the beginning and suddenly changed to 25% MP B from 4.50 min to 6.70 min, and then changed back to 95% MP B for 6.80 – 9.00 min.

2.       In the discussion, the authors claimed that “The present study cannot explain whether the effects observed are solely derived of the presence of D-Pinitol, or if they come from the association of this inositol to glucose and fructose. However, the profile of the effects of the syrup on insulin signaling cascade found on the hippocampus and the hypothalamus were very similar to those described in experimental conditions where D-Pinitol was administered alone”. Any experiments with D-Pinitol alone or D-Pinitol with different compositions of glucose and fructose to confirm the activity of D-Pinitol?

3.       On page 18, the last word in line 565 should be “in” instead of “I”.

Author Response

REFEREE 1

This manuscript presents the exploration of an alternative approach for sweetening using a simplified carob pod-derived syrup consisting of a balanced mixture of fructose, glucose, and a functional natural ingredient (D-Pinitol). The administration of this syrup to healthy volunteers resulted in less persistent glucose excursions, lower insulin response to the hyperglycemia produced by its ingestion and enhanced glucagon/insulin ratio, than that observed after the ingestion of glucose. Daily administration of the syrup to Wistar rats lowered fat depots in the liver, reduced liver glycogen, promoted fat oxidation, and improved glucose handling.

A couple of questions for the authors:

  1. On page 5, line 170-173. The chromatography gradient does not look right to me. Please check carefully. It should start with high percentage of MP A and gradually increase the percentage of MP B till 95% for 6.80-9:00 min. But from the manuscript, it started with 95% of MP B from the beginning and suddenly changed to 25% MP B from 4.50 min to 6.70 min, and then changed back to 95% MP B for 6.80 – 9.00 min.

ANSWER: We appreciate referee’s suggestion. The method is correct. We used a chromatographic column of hydrophilic interaction (X bridge BEH amide, Waters), where the elution process of the compounds in the columns occurs with water. We modified the text to try to clarify the methodology. We started with 95% MP B (0.0-0.5 min), and gradually increased the percentage of MP A to 75% until 4.50 min, this percentage was maintained from 4.50 to 6.70 min, after MP A decreased to 5% at 6.80 min, and this gradient was maintained until 9.00 min.

  1. In the discussion, the authors claimed that “The present study cannot explain whether the effects observed are solely derived of the presence of D-Pinitol, or if they come from the association of this inositol to glucose and fructose. However, the profile of the effects of the syrup on insulin signaling cascade found on the hippocampus and the hypothalamus were very similar to those described in experimental conditions where D-Pinitol was administered alone”. Any experiments with D-Pinitol alone or D-Pinitol with different compositions of glucose and fructose to confirm the activity of D-Pinitol?

ANSWER: we thank the referee for the suggestions. We have incorporated to the manuscript an additional experiment with D-Pinitol(100 mg/kg for 10 days)  where we have measured the same parameters of Table 2. The results obtained support the findings described for the carob syrup on both, fat deposition decrease and urea production. However, we did not see a decrease in hepatic glycogen nor enhancement of plasma ghrelin. This experimental set was originally used for the analysis of D-Pinitol action in the hippocampus. (Medina-Vera D, Navarro JA, Rivera P, et al. d-Pinitol promotes tau dephosphorylation through a cyclin-dependent kinase 5 regulation mechanism: A new potential approach for tauopathies? Br J Pharmacol. 2022 Jun 27. doi: 10.1111/bph.15907), but we kept frozen the . Since this is only a supportive experiment, we have placed it in supplementary materials, Table S2. We have also placed a limitation section where we consider the need of performing additional experiments with glucose + D-Pinitol or fructose + D-Pinitol to fully understand the nature of the action of this polyol.

Supplementary Table S2. Plasma and Liver Biochemistry Parameters after 10 days of drinking water , or  water-diluted D-Pinitol (equivalent to 100 mg/kg b.w. /day of D-Pinitol).

Water

D-Pinitol

N

10

10

Glucose (mg/dl)

247.0 + 63.8

269.4 + 46.3

  Creatinin (mg/dl)

0.57 + 0.38

0.70 + 0.10

Urea (mg/dl)

21.6 + 3.2

41.8 + 2.8 (*)

Bilirubin (mg/dl)

0.10 + 0.09

0.10 + 0.02

Uric Acid (mg/dl)

1.67 + 0.29

1.90 + 0.17

Triglycerides (mg/dl)

146.9 + 30.1

131.1 + 9.39

b-Hydroxy butirate (mg/dl)

1005 + 86

1057 + 90.8

AST (U/L)

152.6 + 46.5

254.6 + 33.18

Insulin (ng/ml))

14.9 + 1.9

14.5 + 0.9

Glucagon/Insulin ratio

26.6 + 4.9

37.3 + 11.9

Leptin (ng/ml)

14.8 + 4.6

12.6 + 3.4

Ghrelin (ng/ml)

0.42 + 0.13

0.53 + 0.14

TBARS (Malonyl dialdehyde, mM)

11.1 + 2.7

8.94 + 1.09

Total Fat in Liver (mg/g)

40.8 + 1.3

30.4 + 6.9 (*)

Liver Glycogen (mg/g)

137.7 + 34.1

107.8 + 10.6

Data are means + Standard Deviation. (*) indicates p<0.05, ANOVA or Kruskal-Wallis test.

AST (Aspartate aminotransferase), TBARS (Tiobarbituric acid reactive species).

  1. On page 18, the last word in line 565 should be “in” instead of “I”.

ANSWER: corrected

Reviewer 2 Report

This manuscript describes an analysis of the short term effects of a new potential sweetener on the glucose metabolism in humans and on glucose and lipid metabolism in rats. It is an interesting concept and certainly timely and important. However, there are a number of fundamental flaws in the human trial that could potentially affect the data and the animal study is of a short duration and will not provide long-term safety data. The animal study also has some changes that could be detrimental in the long run that need addressing. I have put detailed comments below.

Major comments:

1.     The introduction is written with some emotive language, which does move the argument forward but needs to be an active decision by the authors.

2.     Methods, human volunteers: please define normal basal glucose and insulin levels and biochemical parameters (which were considered?) and how was this determined?

3.     Methods, human study: why was a 50 g dose chosen? Wouldn’t it be easier to compare it to the outcomes of a 75g OGTT?

4.     Methods, human study: why were not the same individuals exposed to all three treatments? Given that there is a reasonable variability in the insulin and glucose measures in the volunteers (and thus in HOMA-IR), there is likely to be a reasonably large variability in response to the treatment. A within-subject design would therefore likely have provided more reliable data. I also do not understand why the group sizes are different and of the size that they are. Did the authors do a power calculation to determine the appropriate sample size and how did they do this to get different group sizes?

5.     Methods, effects on hormones: given that the volunteers included women of childbearing age, it is likely that the cycling of sex hormones (and thus the levels of LH and FSH) would be affected by the point in the cycle. How many women were in each of the groups and were they all investigated in the same point in their cycle? Were women with long-term contraception devices excluded?

6.     Methods, rodent studies: why were the rodents only males? How enables this a comparison with the human study? How was sample size determined? While 10 days exposure will give some indication of the longer-term effects, it will of course not provide a true insight into the long-term consequences of the treatment on metabolism. Why was the cut-off of 10 days chosen?

7.     Methods, phosphorylated proteins: given that the animals were fasting when they were sacrificed, the measurements of phosphorylation status will mainly provide an idea of the baseline status. How do the authors think that this will reflect what happens in the post-prandial state which is of major importance for both obesity and diabetes?

8.     Methods, real-time PCR: please provide details on the ng cDNA used and the primer concentrations in this method. Of course, readers can calculate it but you have not provided the volume of the cDNA synthesis reaction and they should not have to do this.

9.     Results: Information on the participants in each of the treatments needs to be provided. Did the interindividual variability in D-pinitol levels correlate with BMI, age or gender?

10.   Results, figure 1E: I do not understand this figure: HOMA-IR is determined by fasting insulin and fasting glucose, not by values during a glucose challenge. Please remove.

11.  Results, supplementary figure 1: this is potentially concerning, with a simultaneous rise in glucagon and insulin in response to the carob sweetener. How do the authors explain this?

12.  Results, supplementary figure 2: the fasting B-OHB levels are quite high for a non-diabetic, healthy population that has been fasting for 12 hours (e.g. 0.28 mM). Can the authors explain this? Did all participants display ketones? Please also keep the units for the BOHB the same between this figure and table 3.

13.  Results, supplementary figure 3: the data should be separated for men and women esp for FSH and LH. The text should read: none of the hormones, not any of the hormones.

14.  Results, figure 2A: please display this in the same way as you have done for the humans, it will make it easier to compare to figure 1 but also to 2B

15.  Results, figure 3: these results would be better represented as individual values with a line through the mean and error bars for the SD. That would show the variability better and with 10 animals per group, this is feasible.

16.  Results, please use the term gluconeogenesis not neoglucogenesis.

17.  Results, table 3: did the animals in both groups consume similar amounts of food?

18.  Results, figures 3 and 4: it is a little surprising that the carob animals increase both lipid oxidation genes and gluconeogenesis genes. It suggests that the metabolism shifts toward energy use, is this reflecting a lower intake of food or something else?

19.  IN the absence of long-term intake data with regard to body composition, hepatic lipid content and also of other organs, it is not yet clear whether this sweetener could be an alternative for use in humans.

20.  Discussion, paragraph 1: this is not really contributing a lot of new information, it mostly restates what is already presented in the introduction. This paragraph can be removed.

21.  Discussion: in this study, only the circulating glucose levels are measured. What happens with circulating fructose levels? This is important given that fructose has independent mechanisms of action as the authors refer to.

22.  Discussion, the increase in ghrelin: ghrelin is of course an appetite stimulating hormone and could lead to increased food intake which would not be good for the prevention of obesity, please discuss.

23.  Discussion: please clarify why decreased hepatic glycogen storage is beneficial? It suggest that the energy is stored in a different way e.g. most likely lipids (not necessarily in the liver).

24.  Discussion, last paragraph: please be careful with overinterpreting the results of the D-pinitol on tau pathologies, that is quite a stretch since the animals used were all very young.

Author Response

REFEREE 2

This manuscript describes an analysis of the short term effects of a new potential sweetener on the glucose metabolism in humans and on glucose and lipid metabolism in rats. It is an interesting concept and certainly timely and important. However, there are a number of fundamental flaws in the human trial that could potentially affect the data and the animal study is of a short duration and will not provide long-term safety data. The animal study also has some changes that could be detrimental in the long run that need addressing. I have put detailed comments below.

ANSWER: We appreciate referee’s comment.  We have addressed the revision taking in consideration these commentaries and including a precise Limitations of the present study section to clarify the aims and achievements and the work pendant to do to complete the characterization of this new syrup. First we must clarify that the study was not intended to be a therapeutic-oriented trial, but a descriptive study on the properties of this type of functional sweeteners. Second, we are aware of the need of addressing vey long term studies in animals for confirming whether this syrup has not toxic effects despite the fructose it contains. However acute /short-term administration actions described in the present study granted this future studies since the results obtained indicated neither liver nor kidney toxicity.

 Major comments:

  1. The introduction is written with some emotive language, which does move the argument forward but needs to be an active decision by the authors.

ANSWER: we have modified the introduction to avoid emotive expressions as suggested.

  1. Methods, human volunteers: please define normal basal glucose and insulin levels and biochemical parameters (which were considered?) and how was this determined?

ANSWER: Since the study was not oriented to address any clinical condition, recruitment criteria included a) the presence of a baseline capillary blood glucose  < 5.6 nmol/L (100 mg/mL) measured with a glucose oxidase method after overnight fasting,  b) the absence of obesity (BMI > 30) and the absence of a diagnosed/treated metabolic disease. Subjects were interviewed for present or past diagnosis of diabetes (Glucose >7.8 mmol/L after a standard glucose load), hypertryglyceridemia or hypercholesterolemia under treatment, as well as for a clinical record of past endocrine disorders including thyroid gland dysfunction or treatment with glucocorticoids. Reference values for normal insulin concentrations determined by ELISA after overnight fasting were considered > 25mIU/L. This information has been added to the manuscript on the recruitment section.

  1. Methods, human study: why was a 50 g dose chosen? Wouldn’t it be easier to compare it to the outcomes of a 75g OGTT?

ANSWER: We thank the referee for the comment since we did not explain correctly the methodology. We did not intend to evaluate a standard OGTT but to obtain the glycemic index of the  carob syrup, using a standard procedure. The glycemic index value of a food product is determined by feeding healthy people a portion of the food (in this case the carob syrup and the agave sweetener) containing 50 grams of digestible carbohydrates and then measuring its effect on their blood glucose levels over the next 2 hours. Then the area under the two-hour blood glucose response (glucose AUC) graph each person is measured. Data are compared with the AUC obtained after consumption of an equal-carbohydrate portion of pure glucose (50 grams) as a reference meal. By error we neither explain this, nor placed the calculation of the glycemic index. The description and results are now placed in the respective sections of the manuscript.

  1. Methods, human study: why were not the same individuals exposed to all three treatments? Given that there is a reasonable variability in the insulin and glucose measures in the volunteers (and thus in HOMA-IR), there is likely to be a reasonably large variability in response to the treatment. A within-subject design would therefore likely have provided more reliable data. I also do not understand why the group sizes are different and of the size that they are. Did the authors do a power calculation to determine the appropriate sample size and how did they do this to get different group sizes?

ANSWER: We agree with the referee that a within subject design is a better option than a pure in between one. However, because of the need of repeated blood drawing, and the unknown short-term repercussions derived of the exposure to the D-Pinitol present in the syrup, we decided to select the in between subject design. This is a limitation of the study that we have described in the correspondent section.

Concerning the sample size, we calculated it using the Gpower program, version 3.1.9.2. The main variable of the study was the area under the curve (AUC) derived of plasma glucose excursions after oral intake of either, a glucose solution, carob syrup or agave syrup. Considering that compared with the standard glucose solution used (50 gr), carob syrup will have around 25 grams (50%), and agave syrup around 20 grams (1/5) of  glucose content, a size of effect of 1.6 for glucose and  a size of effect of 2 was considered for carob syrup an agave syrup respectively. That gives a sample size of 10 subjects per group for comparing glucose versus carob syrup, and a size sample of 7 for comparing glucose versus agave syrup. That means a total of 27 subjects.  However, after the recruitment period was closed, we only recruited 23 subjects that were allocated to the 3 groups, in a proportional distribution with respect to the calculated sample size. This information has been incorporated to the manuscript.

  1. Methods, effects on hormones: given that the volunteers included women of childbearing age, it is likely that the cycling of sex hormones (and thus the levels of LH and FSH) would be affected by the point in the cycle. How many women were in each of the groups and were they all investigated in the same point in their cycle? Were women with long-term contraception devices excluded?

ANSWER: We appreciate referee’s comments. In the Carob Syrup group there were 9 subjects, 4 males and 5 females; in the glucose group 5 males and 3 females; and finally, in the agave gropu there were 4 males and 2 females. Subjects were randomly cycling and we did not record the expected phase of the reproductive cycle. Two of the females displayed high LH/FSH and a rise of PRL that indicated they were on the periovulatory phase of the eproductive cycle, and were initially excluded from the hormonal analysis reported in the first version of the study. However, following referee’s suggestion, we have now included all data, and separated females from males. No one of the females recruited used contraceptive measures (neither devices nor medication). Final analysis revealed that the only source of hormonal differences was sex, and that D-Pinitol is not affecting these hormones. This analysis is now placed in the results section and a new graph is allocated as Supplementary graph S3.

  1. Methods, rodent studies: why were the rodents only males? How enables this a comparison with the human study? How was sample size determined? While 10 days exposure will give some indication of the longer-term effects, it will of course not provide a true insight into the long-term consequences of the treatment on metabolism. Why was the cut-off of 10 days chosen?

ANSWER: The referee is right with respect to female studies in animal models that will be addressed in future studies. We did not use female animals in this short term study until understanding the acute effects of D-Pinitol in hormonal surges along the estrous cycle that might eventually have impact on metabolism and fat depot in the liver. This is a limitation that is included in the new version of the study. With respect to the number of animal we used a standard size of 8-10 animals per group because they are inbred strains (Wistar rats) with a very close and stable genetics that allows lower variance on experimental results. We selected this option since we followed the 3R recommendation of the EU that promotes the reduction of experimental animals.  The 10 days chosen was selected precisely for addressing potential early toxic events, and we agree that long-term studies, comparing carob syrup with agave or high fructose corn syrup, will be necessary to discard any long-term hepatic damage attributable to the fructose content in the carob syrup.

  1. Methods, phosphorylated proteins: given that the animals were fasting when they were sacrificed, the measurements of phosphorylation status will mainly provide an idea of the baseline status. How do the authors think that this will reflect what happens in the post-prandial state which is of major importance for both obesity and diabetes?

ANSWER: This is a very interesting question to be studied in animal models, both, control and diabetic. However, the proposed experimental research was far beyond of the initial aims of the present study and will be addressed in future studies. We have mentioned this need in the discussion.

  1. Methods, real-time PCR: please provide details on the ng cDNA used and the primer concentrations in this method. Of course, readers can calculate it but you have not provided the volume of the cDNA synthesis reaction and they should not have to do this.

ANSWER: We have provided the detailed information requested by the referee.

  1. Results: Information on the participants in each of the treatments needs to be provided. Did the interindividual variability in D-pinitol levels correlate with BMI, age or gender?

ANSWER: There is a table (Table 1)  with the information concerning the recruited participants. Subsequent analysis were clearly depicted for the different experimental groups. We have performed correlation analysis of Peak concentrations of D-Pinitol versus age, BMI and analysis of gender differences. D-Pinitol correlates positively  with age (r2=0.45, p (two tailed) = 0.047). There were no correlations with D-Pinitol with BMI (r2=0.44, p (two tailed) = 0.05). Mean peak plasma D-Pinitol values were similar in females (1527 + 589 ng/ml) than in male subjects (1562 + 178 ng/ml). This information can be found now in the manuscript.

  1. Results, figure 1E: I do not understand this figure: HOMA-IR is determined by fasting insulin and fasting glucose, not by values during a glucose challenge. Please remove.

ANSWER: We have removed this figure, as suggested. It has been substituted by that of the plasma fructose levels.

  1. Results, supplementary figure 1: this is potentially concerning, with a simultaneous rise in glucagon and insulin in response to the carob sweetener. How do the authors explain this?

ANSWER: Preclinical experiments in animal models published previously by the research group (Navarro et al., Nutrients. 2020 Jul 8;12(7):2030.) suggested that D-Pinitol inhibits insulin secretion from pancreatic islets, probably through enhanced ghrelin secretion and/or the inhibition of ERK1-2 signaling in the beta cells. This effect is accompanied by a rise of glucagon that sustains glycaemia, promoting gluconeogenesis from the liver. The final result is that glycaemia is sustained despite the drop on insulin secretion.  The precise mechanism by which glucagon release is promoted after D-Pinitol administration remains to be determined in experimental rodents. However, our findings suggest that this mechanism might be also present in humans. This hypothesis will require further investigation.

  1. Results, supplementary figure 2: the fasting B-OHB levels are quite high for a non-diabetic, healthy population that has been fasting for 12 hours (e.g. 0.28 mM). Can the authors explain this? Did all participants display ketones? Please also keep the units for the BOHB the same between this figure and table 3.

ANSWER: We agree with the referee with respect to the comment on the concentrations of B-OHB that are in the high range of values for fasting humans. Thus we have included ketone measurements that clearly show that the experimental subjects did not display elevated ketone concentrations in plasma, despite fasting for 12 hours. This graph is located as supplementary graph S2.

  1. Results, supplementary figure 3: the data should be separated for men and women esp for FSH and LH. The text should read: none of the hormones, not any of the hormones.

        ANSWER:  We have done the analysis as suggested. There were 2 women with high levels of FSH and LH, and a partial elevation of prolactin, suggesting that they were in periovulatoy period. These 2 subjects were initially removed from the analysis, and now they are included. Statistically analysis confirms that D-Pinitol was not affecting plasma pituitary hormone concentrations, and that the only source of variation is sex. These new graphs are now located as supplementary figures S3.

  1. Results, figure 2A: please display this in the same way as you have done for the humans, it will make it easier to compare to figure 1 but also to 2B

        ANSWER: We have modified the graph accordingly

  1. Results, figure 3: these results would be better represented as individual values with a line through the mean and error bars for the SD. That would show the variability better and with 10 animals per group, this is feasible.

        ANSWER: We have modified PCR graphs accordingly.

  1. Results, please use the term gluconeogenesis not neoglucogenesis.

        ANSWER: corrected

  1. Results, table 3: did the animals in both groups consume similar amounts of food?

        ANSWER: We did not measure food intake, since previous studies performed at the laboratory showed that the inclusion of the syrup in the drinking water did not modify food intake. Moreover, acute intragastric administration of a carob syrup dose equivalent to 100 mg/kg of D-Pinitol did not modify food intake in 18 h food deprived animals. This experiment has been placed as supplementary material figure S4.

  1. Results, figures 3 and 4: it is a little surprising that the carob animals increase both lipid oxidation genes and gluconeogenesis genes. It suggests that the metabolism shifts toward energy use, is this reflecting a lower intake of food or something else?

        ANSWER: Please see below an integrated response to this question together with those raised on point 23.

  1. IN the absence of long-term intake data with regard to body composition, hepatic lipid content and also of other organs, it is not yet clear whether this sweetener could be an alternative for use in humans.

        ANSWER: This study is not intended as a conclusive report for determining its use as a potential sweetener in diabesity, but an initial step for characterizing the physiological responses of a sweetener containing glucose, fructose and D-Pinitol. We agree with the referee that future studies are needed to fully confirm its utility, once we have described the acute and subacute administration effects.

  1. Discussion, paragraph 1: this is not really contributing a lot of new information, it mostly restates what is already presented in the introduction. This paragraph can be removed.

        ANSWER: Done.

  1. Discussion: in this study, only the circulating glucose levels are measured. What happens with circulating fructose levels? This is important given that fructose has independent mechanisms of action as the authors refer to.

ANSWER: We have measured fructose levels and discussed its relevance as suggested. A new graph with fructosemia values is displayed as figure 1E. Methods used are included now in the manuscript, as well as a paragraph in the discussion section.

  1. Discussion, the increase in ghrelin: ghrelin is of course an appetite stimulating hormone and could lead to increased food intake which would not be good for the prevention of obesity, please discuss.

        ANSWER: Ghrelin has multiple actions in the body, including the control of appetite. In our experimental model we have observed that feeding is not affected despite the rise in ghrelin in animal models. We have no information in human subjects, so we have discussed the convenience to address an experimental study on appetite after acute and chronic administration of carob syrup to healthy human subjects. This is a necessary step before future clinical trials on obesity and diabetes type 2.

  1. Discussion: please clarify why decreased hepatic glycogen storage is beneficial? It suggest that the energy is stored in a different way e.g. most likely lipids (not necessarily in the liver).

        ANSWER: What we have found is that carob syrup exposure redirects liver metabolism towards glucose export and lipid oxidation. We attribute these effect to D-Pinitol, since previous studies with this inositol (Navarro et al., Nutrients. 2020 Jul 8;12(7):2030.) produces the same set of effects. In this study we observed a decreased in insulin secretion associated to a rise of ghrelin and glucagon. Since D-Pinitol is able tof activating both glucose uptake in muscle cells (Bates SH et al.  Br J Pharmacol. 2000 Aug;130(8):1944-8) and canonical insulin signaling in the hypothalamus (Medina-Vera et al. Nutrients. 2021 Jun 30;13(7):2268.) our interpretation is that D-Pinitol favours glucose transfer from liver to other tissues, including muscle. We do not know the impact of D-Pinitol ( given alone or under the carob syrup format) on both adipose tissue function and energy expenditure, so future studies must address this important aspects to understand the pharmacological profile of this inositol. We have included these reflections in the discussion.

  1. Discussion, last paragraph: please be careful with overinterpreting the results of the D-pinitol on tau pathologies, that is quite a stretch since the animals used were all very young.

ANSWER: We have recently published a manuscript where we demonstrate that chronic administration of D-Pinitol in normal and Alzheimer’ Disease human mutation-bearing animals reduce tau phosphorylation through a CDK5-dependent mechanism. (See Medina-Vera D, Navarro JA, Rivera P, et al. d-Pinitol promotes tau dephosphorylation through a cyclin-dependent kinase 5 regulation mechanism: A new potential approach for tauopathies? Br J Pharmacol. 2022 Jun 27. doi: 10.1111/bph.15907). It is true that we have not tested the syrup in the animal models of AD, but the results found in the present study are encouraging. We have modified the last paragraph to indicate this possibility, as well as the need of addressing new experiments with the carob syrup on tau pathologies.

Round 2

Reviewer 2 Report

thank you to the authors for their extensive revision of the manuscript. I have no further comments.